# Silicone Induced Granuloma of Breast Implant Capsule (SIGBIC) diagnosis: Breast Magnetic Resonance (BMR) sensitivity to detect silicone bleeding

Eduardo de Faria Castro Fleury [1,2]*

1 Centro Universitário São Camilo, Curso de Medicina – São Paulo, São Paulo Brazil, 2 Instituto Brasileiro de Controle do Câncer – São Paulo, São Paulo, Brazil

* edufleury@hotmail.com

**Data Availability Statement:** Protocol is available at protocols.io (DOI: http://dx.doi.org/10.17504/protocols.io.bfn4jmgw). Data are available at Dryad (DOI: 10.5061/dryad.t76hdr7z5).

## Abstract

### Objective

To evaluate the sensitivity (S) of BMRI to detect silicone gel bleeding in a prospective observational study, including consecutive patients referred for BMRI scan.

### Methods

From January 2017 to March 2018, we evaluated patients with breast implants referred for BMRI in a prospective observational study. For SIGBIC diagnosis, we adopted three new original imaging features: black drop signal; T2* hyper signal mass; and delayed contrast enhancement, considered as irrevocable signs to detect gel bleeding (GB). Histology confirmed the presence of a silicone corpuscle in breast implant capsular specimens. The accuracy of BMRI SIGBIC findings to predict GB was determined. We also compared SIGBIC diagnosis criteria to those features proposed by the BI-RADS léxicon, considered as equivocal findings.

### Results

208 patients had SIGBIC diagnosis at BMRI, and the histology confirmed GB in all cases. There were no false-positive results. Compared to the BI-RADS equivocal findings (S = 0.74), SIGBIC criteria had better sensitivity for GB diagnosis.

### Conclusion

SIGBIC diagnosis has high sensitivity to predict GB by the three irrevocable BMRI features described by the authors. We suppose GB is underdiagnosed in clinical practice by BI-RADS features.

### Trial certification

Study protocol: Plataforma Brasil **CAAE**: 77215317.0.0000.0072.

**Funding:** There is no grant founding/ financial support for this manuscript.

**Competing interests:** The authors have declared that no competing interests exist.

**Abbreviations:** BIA-ALCL, breast implant-associated anaplastic large cell lymphoma; BMRI, breast magnetic resonance imaging; GB, gel bleeding; SIGBIC, silicone-induced granuloma of breast implant capsule; SIIS, silicone implant incompatibility syndrome.

## Introduction

Over the last two years, an increasing number of studies have reported complications related to breast silicone implants [1–4]. These complications are commonly associated with the incidence of breast implant-associated anaplastic large cell lymphoma (BIA-ALCL) [5–12]; however, the onset of this pathology and its trigger point is yet to be elucidated.

Interestingly, many of these changes have been reported despite the macroscopic integrity of breast implants. The main clinical complications related to breast implants are breast stiffness, late seroma, lymph node enlargement, and silicone migration to distant organs. Also, clinical signs related to autoimmune reactions have been reported, which highlight the silicone implant incompatibility syndrome (SIIS) [13].

Gel bleeding is not virtually ignored, but, as far as we know, there are indeed no demonstrations that can determine any of the known complications.

Recently we have described a new radiological finding, silicone-induced granuloma of breast implant capsule (SIGBIC) [14], defined as granulation tissue formed from a reaction between the breast implant fibrous capsule and free silicone corpuscle due to bleeding of the intact breast implant. We have described 3 BMRI features that are irrevocable for this diagnosis: 1. black-drop signal; 2. mass with hyper signal at T2-weighted sequences; and 3. late contrast enhancement [15]. Besides, we described the physiopathology of the disease [16]. (Fig 1).

This study aimed to determine the ability of BMRI to predict GB according to SIGBIC diagnosis with the three irrevocable features. We also compared the performance of SIGBIC to diagnose GB to the descriptors proposed by the BI-RADS lexicon. This manuscript is the first prospective study to determine its prevalence in clinical practice.

## Materials and methods

This study was a prospective observational study conducted at a single academic institution with subspecialty training in breast imaging from March 2017 to August 2018 following local ethics committee regulations. We obtained free, written, informed consent from all patients.

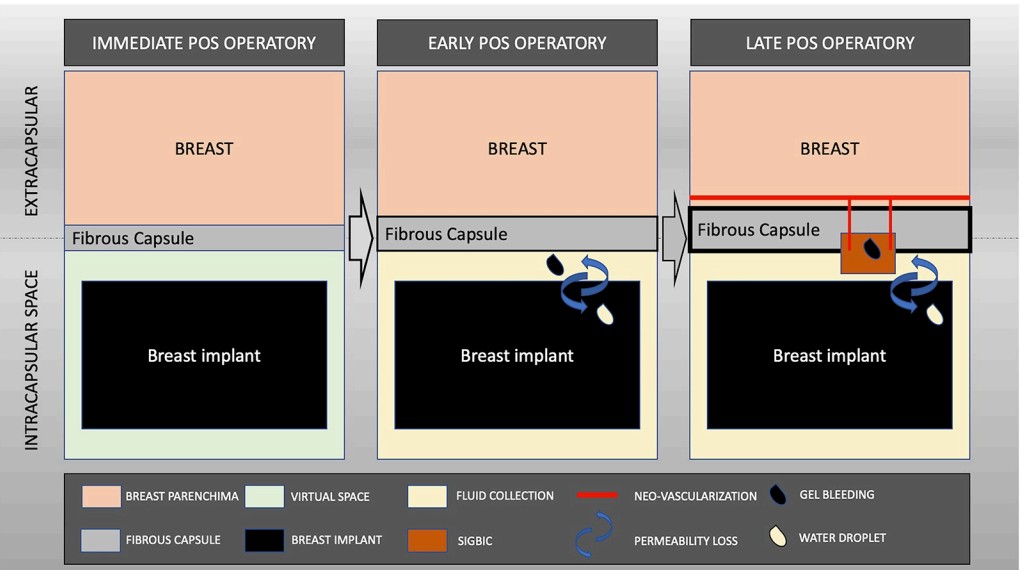

**Fig 1. Scheme of Silicone Induced Granuloma of Breast Implant Capsule (SIGBIC) development and its imaging findings.**

We evaluated patients referred for BMRI. All patients who had breast implants were included in the study. The exclusion criteria were patients with previous BMRI scans during the study period to avoid duplicity of the results, technically inappropriate scans, and scans without contrast injection. The research was deposit in protocols.io as http://dx.doi.org/10.17504/protocols.io.bfn4jmgw. The study was approved by the institutional ethical committee (Instituto Brasileiro de Controle do Câncer). Inform consent was obtained from all of the participants. Study protocol: Plataforma Brasil CAAE: 77215317.0.0000.0072.

We used a standard BMRI factory protocol for a 1.5-T imager (Magnetom Aera; Siemens Healthcare) with a dedicated eight-channel bilateral breast coil in the axial orientation. The acquired sequences were: 1. Axial T2-weighted fast spin-echo; 2. Sagittal Proton-density weighted; 3. Sagittal T2-weighted silicone selective that enhance the silicone signal; 4. Sagittal T2-weighted sequence with silicone suppression that extract the silicone signal; and 5. 4 following dynamic contrast sequences with fat suppression and reconstruction with subtraction.

One radiologist with 18 years of experience in BMRI read all the acquired images.

Three novelty imaging features were adopted for SIGBIC diagnosis:

a.  black drop signal: a marked focus of low signal at T1 sequences in the fibrous capsule, and a silicone signal focus could be associated in silicone sensitive sequences, without enhancement in post-contrast sequences;

b.  mass with hyper signal at T2 weighted sequence: an intracapsular mass that could misdiagnose as seroma

c.  late contrast enhancement: the mass (b) shows late contrast enhancement at the late phase.

SIGBIC diagnosis was only determined when all the three irrevocable features were present.

Also, and for comparison purposes, we described nine established BMRI findings according to the BI-RADS lexicon, considered as equivocal signs:

1.  Capsular contracture: increase of the anteroposterior diameter of the implant with fibrous capsule thickening and contrast enhancement (Figs 2,3, 4 and 5).

2.  Intracapsular rupture: discontinuity of the breast implant surface, often characterized by subcapsular line, keyhole sign, teardrop sign, and linguine sign restricted to intracapsular space.

3.  Extracapsular rupture: discontinuity of the fibrous capsule with extravasation of the internal silicone contents.

4.  Water droplets: foci of water signal inside the intact breast implant inferring loss of surface permeability (Fig 2).

5.  Implant Rotation: posterior surface of the breast implant displacement, often identified by the ectopic position of the implant seal (Fig 3).

6.  Extracapsular Siliconoma: extracapsular free silicone without signs of fibrous capsule rupture.

7.  Enlarged intramammary lymph node (EILN): enlarged pericapsular intramammary lymph node (Fig 4). We used two criteria to determine EILN: smaller diameter greater than 0.5cm (perpendicular axis) and thickening of the lymph node cortex with reduced fatty hilum.

8.  Pericapsular edema: non-mass pericapsular enhancement (Fig 5).

9.  Intracapsular seroma: collection inside the fibrous capsule (Figs 2 and 3).

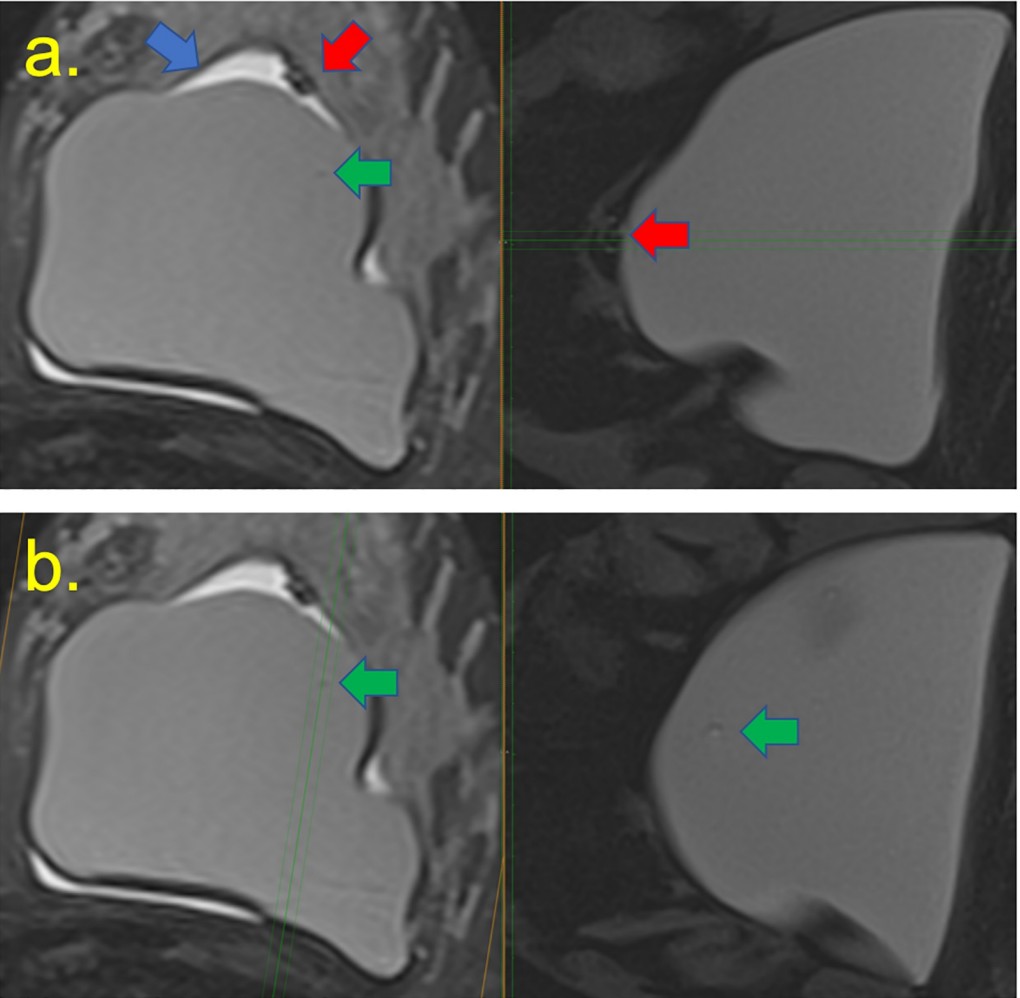

**Fig 2. A 45-year-old woman 6 years after implant placement with left breast stiffness T2-weighted and silicone-sensitive images on the right and left, respectively.** Blue and red arrows demonstrate an intracapsular seroma and mass that includes free silicone. (a) Intact breast implant. (b) Green arrow demonstrates water droplets.

Patients diagnosed with SIGBIC at BMRI underwent a second-look ultrasound scan using a device with a 7.5–14 MHz multi-frequency probe (Aplio 300, Toshiba). All the patients who met the SIGBIC BMRI diagnostic criteria and were symptomatic underwent US-guided percutaneous core-biopsy or directed for surgical capsulectomy. Symptomatic criteria for biopsy were: breast enlargement refractory to clinical treatment, local inflammatory signs, and ordinary daily activities limitation. We opted for the percutaneous biopsy in patients who have easy access to performing the procedures. In patients where the lesions were very deep or posterior to the breast implants, capsulectomy has opted. We performed percutaneous biopsy using a 14-G needle attached to an automatic biopsy gun. At least three samples were collected. The remaining BMRI equivocal findings were followed up.

The specimens from biopsies or surgical procedures were evaluated in the same institution, where a diagnosis of SIGBIC was confirmed when the silicone particle was observed at microscopy, as described in a previous study. (Fig 6).

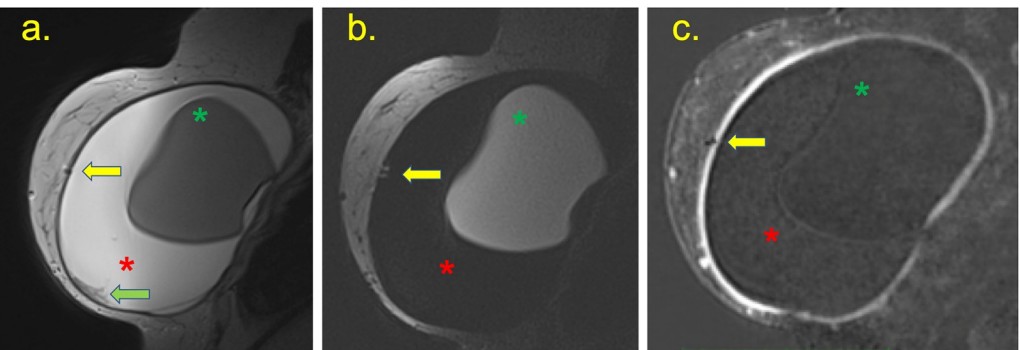

**Fig 3. A 30-year-old woman who underwent an aesthetic procedure for silicone prosthesis four years ago, 1 week before an increase in volume and inflammation of the right breast.** (a) STIR sequencing presented a massive intracapsular collection (red asterisk) with intact and rotated breast implant (green asterisk), and free silicone in the fibrous capsule forming the granuloma (yellow arrow). (b) Identical results were found with T2W silicone-sensitive, and after the use of contrast. (c) Findings were compatible with the black drop signal. Intact breast implant.

The main objective of this study was to determine the ability of BMRI to predict gel bleeding in patients with SIGBIC diagnosis compared with histopathology as the gold standard. As secondary objectives, we established what equivocal BMRI features could be related to GB. In this context, a univariate analysis was performed through Chi-Square tests [17] and Fisher's Exact Test [18] and the test of Mann-Whitney [19] for the case of categorical variables [20]. The approach used was not the automatic Stepwise. Through the univariate analysis, we selected the potential predictors for the response variable, considering a level of significance equal to 25%, according to Kennedy and Bancroft criteria [21]. The multivariate Logistic Regression model was only used for patients with histopathological proved SIGBIC in order to determine diagnosis ability using ONLY the established BI-RADS lexicon BMRI features.

We also evaluated, in order to verify if the adjusted models were adequate and if they had functional predictive capacity, some measures of quality of adjustment using only the association of the main equivocal features (BI-RADS lexicon) for GB diagnosis: Pseudo $R^2$ [22], AUC (area under the ROC curve), Sensitivity, Specificity, VPP, and the Hosmer-Lemeshow test [18]. The software used in the analysis was R (version 3.5.2).

## Results

We performed 2290 consecutive BMRI from March 2017 to August 2018. Of these, 736 patients had breast implants. Fifty-six patients were excluded from the study, 48 because they refused the injection of contrast medium, six because the presence of motion artifacts or difficulty in performing the silicone sequences, and two dues to previous BMRI scans during the study period. We also excluded 472 patients without the three diagnostic criteria for SIGBIC (equivocal features).

The remaining 208 patients with SIGBIC diagnosis by BMRI (irrevocable signs) were included in the study. All 208 patients with the irrevocable findings had GB confirmed by silicone corpuscles in histopathology. Histological specimens of 40 (19.9%) cases were obtained by surgical capsulectomy and 168 (80.1%) from percutaneous breast biopsy. There were no false-positive results of GB adopting the SIGBIC irrevocable criteria. All capsular specimens showed silicone particles associated with the inflammatory response at histology, and all patients submitted to surgical capsulectomy have no apparent signs of implant rupture. We did not report any complications related to percutaneous breast biopsy.

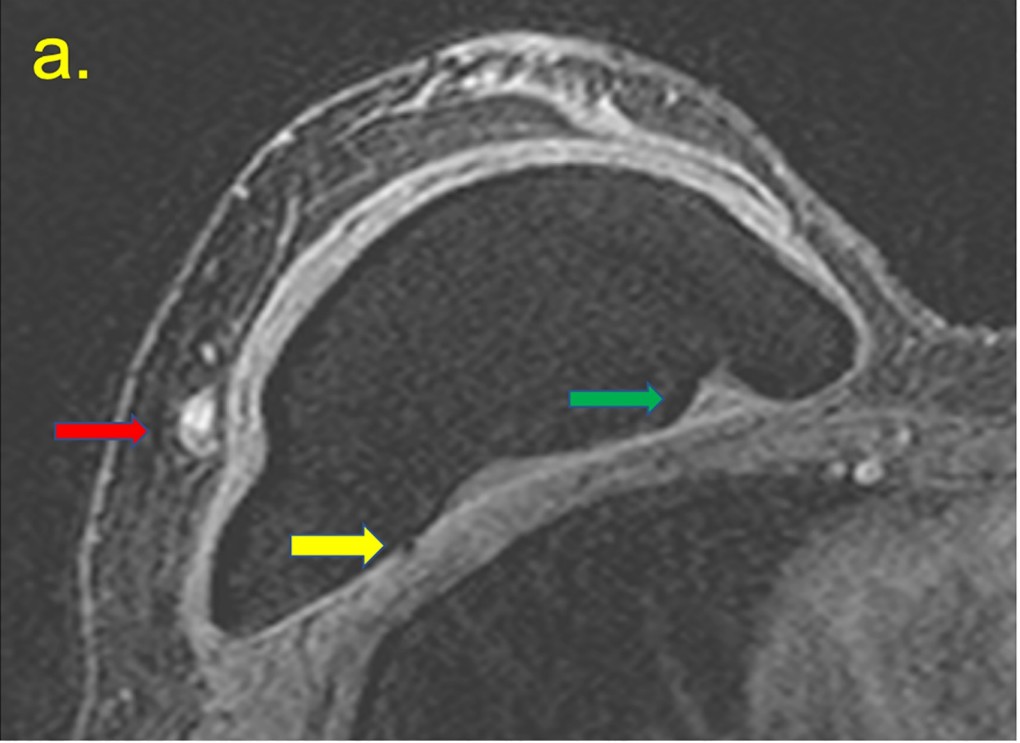

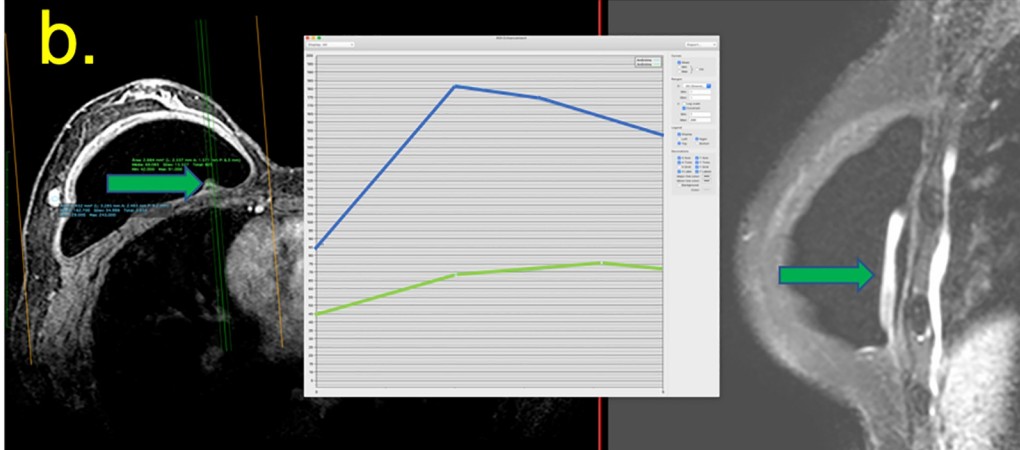

**Fig 4. A 62-year-old patient underwent right breast repair surgery three years ago, followed by treatment with radiotherapy sessions.** Arthralgia had been reported for 3 months. (a) Intracapsular masses are shown in the pre-contrast sequence (green arrow), with enlarged intramammary lymph nodes (red arrow), and black-drop signal (yellow arrow). (b) The mass has late contrast enhancement (green line at the graph) compared to the lymph node early enhancement (blue line). The mass has a high signal at T2 weighted sequence, and the implant was intact.

The results for the equivocal features, described by the BI-RADS lexicon to diagnose GB were as follows:

## Univariate analysis

Table 1 shows the univariate analysis using the Chi-Square test [17] for categorical explanatory variables with expected frequencies higher than 5 in all class and Fisher's exact test [17]. The equivocal features with to predict GB diagnosis were: water droplet, enlarged intramammary

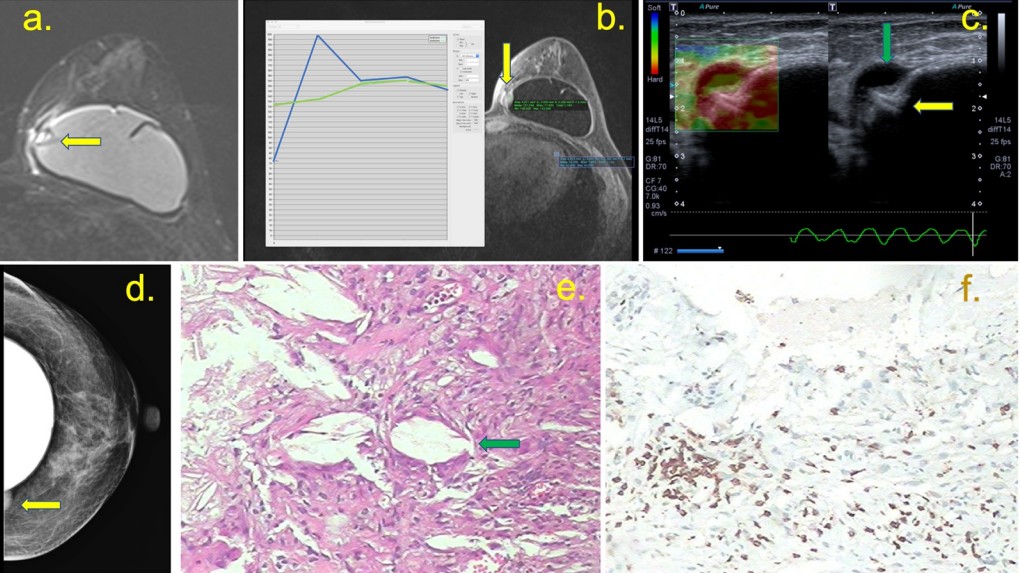

**Fig 5. (a, b, c, d, e and f) A 37-year-old patient with a breast implant for 3 years, who showed a palpable mass in the medial quadrants of the left breast for 3 weeks.** (a) T2W fat-suppressed imaging showing a capsular contracture with a hypersignal mass in the medial quadrant and pericapsular edema (yellow arrow). The post-contrast sequence shows the late enhancement of this mass (green curve) compared with the heart (blue curve). (b) A pericapsular enhancement is shown (yellow arrow). (c) Ultrasound with elastography shows siliconoma at the site *via* BMRI, characterized by a hard mass at and snowstorm artifacts (yellow arrow), with a small collection at the periphery (green arrow). (d) Mammography shows a pericapsular mass in the medial quadrant (yellow arrow). (e) Histology showing free silicone particles and inflammatory cells (magnification, 50×). Immunohistochemistry for T lymphocyte (CD3) positive. Intact breast implant.

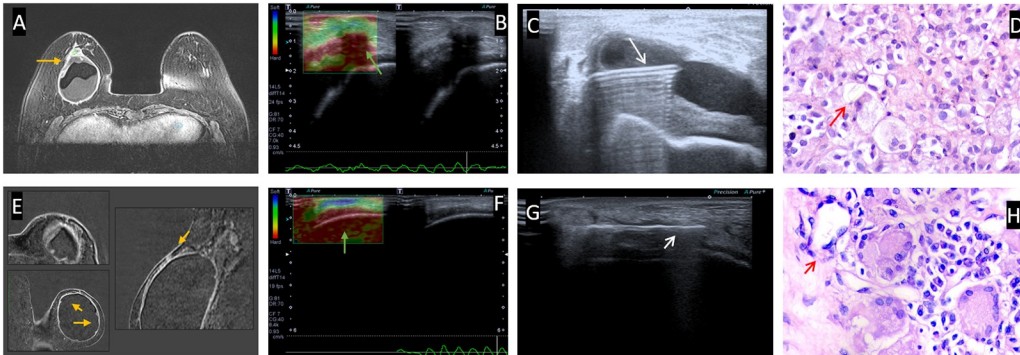

**Fig 6. Examples of percutaneous breast biopsy in 2 patients.** Patient 1 (A, B, C and B). A 49-year-old patient with aesthetic breast implants for 5 years. For 1 month she related breast enlargement with local inflammatory signs on her right breast; and Patient 2 (e, f, g and h). A 68-year-old patient who underwent adenomastectomy and breast reconstruction with breast implants followed to radiotherapy sessions for 1 year. She reported breast stiffness and peripheral arthralgia. At BMRI (A and E), findings suggestive of SIGBIC pointed by the yellow arrow. Ultrasound elastography (B and F) showing a hard mass at the fibrous capsule (green arrow). Biopsy of the mass (C and G) using first a fine needle aspiration to collect the intracapsular seroma (C) and core biopsy of the mass (G) represented by the white arrow. Histological specimens confirming silicone granuloma in red arrow (D,H). In patient 1 we can see intracellular silicone in histocytes (foamy histiocytes) in figure D, and extracellular silicone with giant cell in figure H. Often, intracapsular collection is reported in the presence of intracellular silicone (C,D).

**Table 1. Univariate analysis of breast MRI features.**

| Variables | Category | SIGBIC (+) | | P-value |
|---|---|---|---|---|
| | | N (208) | % | |
| Capsular Contracture | (-) | 0 | 0 | <**0.001**[2] |
| | (+) | 208 | 30.6% | |
| Intracapsular Rupture | (-) | 207 | 100.0% | - |
| | (+) | | | |
| Extracapsular Rupture | (-) | 208 | 30.8% | 0.319[2] |
| | (+) | 0 | 0.00% | |
| Water droplet | (-) | 142 | 25.7% | <**0.001**[1] |
| | (+) | 66 | 51.97% | |
| Implant Rotation | (-) | 198 | 31.2% | 0.274[1] |
| | (+) | 10 | 22.2% | |
| Siliconoma Extracapsular | (-) | 207 | 30.6% | 1.000[2] |
| | (+) | 1 | 25.0% | |
| Enlarged Intramammary Lymph Node (EILN) | (-) | 179 | 28.3% | <**0.001**[1] |
| | (+) | 29 | 60.4% | |
| Pericapsular edema | (-) | 179 | 27.9% | <**0.001**[1] |
| | (+) | 29 | 74.4% | |
| Intracapsular seroma | (-) | 166 | 27.7% | <**0.001**[1] |
| | (+) | 42 | 51.2% | |

[1]Chi-Square test.

[2]Fisher's Exact test.

[3]Mann-Whitney test

SIGBIC: silicone induced granuloma of breast implant capsule

lymph node, pericapsular edema, and intracapsular seroma. We adopted a statistical significance of p<0.001. The variable capsular contracture did not participate in this analysis since it was present in all patients.

## Multivariable analysis

A multivariate model of Logistic Regression [17] was adjusted from the variables selected in the univariate analysis, and for this model, the Backward method was applied for the final selection of the variables, considering a significance level of 5%.

Table 2 presents the final multivariate model for the GB variable with the equivocal features described by the BI-RDADS lexicon. When positive an individual has the chance to have GB multiplied by 2.8 [1.8; 4.4] in the presence of water droplet, 3.07 [1.5; 6.1] in the presence of enlarged intramammary lymph node, 5.02 [2.3; 11.1] in the presence of pericapsular edema, and 2.4 [1.4; 4.1] for intracapsular seroma.

Table 3 shows the negative predictive value (NPV), Hosmer-Lemeshow test, and Pseudo-R2 test for the determination of GB for the final model adopting the most statistically significant equivocal BMRI features. It should be noted that: the area under the ROC curve (AUC) of the final model was 0.750, and the Pseudo-R2 was 24.79%. The Hosmer-Lemeshow test for the final model indicated that the fit of the model was adequate (p-value = 0.795).

## Discussion

Currently, silicone bleeding is virtually ignored as a possible trigger factor for complications related to silicone implants. Most articles illustrate silicone bleeding as a rare event and of little

**Table 2. Multivariate analysis associating BMRI variables with and without gel bleeding.**

| Variables | | O.R. | C.I. (95%) | P-value |
|---|---|---|---|---|
| Water droplet | (-) | 1.0 | - | - |
| | (+) | 2.8 | [1.8; 4.4] | **<0.001** |
| Enlarged Intramammary Lymph Node | (-) | 1.0 | - | - |
| | (+) | 3.1 | [1.5; 6.1] | **0.001** |
| Pericapsular edema | (-) | 1.0 | - | - |
| | (+) | 5.0 | [2.3; 11.1] | **<0.001** |
| Intracapsular Seroma | (-) | 1.0 | - | - |
| | (+) | 2.4 | [1.4; 4.1] | **0.001** |

O.R. odds ratio

C.I. confidence interval

clinical relevance. They justify this statement due to the evolution of silicone breast implants, especially with the use of cohesive gel.

Since the description of SIGBIC by our group, we observed an increasing prevalence of SIGBIC in our clinical practice; therefore, we conducted this prospective study to determine BMRI ability to predict GB in patients with SIGBIC diagnosis. We also intended to assess the association between BMRI equivocal features described by the latest BMRI BI-RADS lexicon and SIGBIC in predicting GB.

Reported breast implant-associated complications have recently been increasing, including BIA-ALCL and Breast Implant Illness in academia, mainstream, and social media. Despite the reported severity associated with anaplastic lymphoma, the number of cases described in the literature remains small, with <750 cases reported, minimizing its relevance [11,12]. Furthermore, the trigger point to develop this pathology remains unclear.

We recently described the radiological findings of a granuloma developed in the intracapsular compartment of breast implants that was formed by an immune response of the fibrous capsule to the free silicone particles bleeding from intact breast implants [16]. After a certain time or when subjected to stress situations (like heat and trauma), all breast implants may alter the surface permeability and may present gel bleeding of the internal contents or leakage of the shell [1]. Both the silicone and saline implant have the substance polydimethilsilixonase (PDMS) as a shell component. Some researchers speculated that this substance could elicit an immune response when in contact with the fibrous capsule. Some patients have a higher risk of developing an immune response, for example, those with SIIS. SIIS clinical manifestations may vary and depend on the intensity of the inflammatory reaction of the host. [16].

**Table 3. Quality statistics of the multivariate model for GB diagnosis adopting water droplets, enlarged intramammary lymph nodes, pericapsular edema and intracapsular seroma.**

| SEN | SPE | PPV | NPV | Accuracy | AUC | Hosmer-Lemeshow | Pseudo-R$^2$ |
|---|---|---|---|---|---|---|---|
| 0.740 | 0.638 | 0.474 | 0.848 | 0.669 | 0.750 | 0.795 | 24.79% |

SEN: sensitivity

SPE: specificity

PPV: positive predictive value

NPV: negative predictive value

AUC: area under curve

In our study, 30.6% of patients with breast implants referred for BMRI scan at our facility fulfill the irrevocable BMRI criteria for SIGBIC diagnosis. All these cases were confirmed by histopathology by the presence of silicone corpuscles.

Due to the novelty of the study, we chose very restrictive criteria for the SIGBIC diagnosis. We aimed to avoid false-positive results in this context. The black-drop signal consists of a giant cell reaction to a foreign body in the fibrous capsule. The mass with hyper signal at the T2-weighted sequence is the development of granulation tissue in the contacted area between the silicone corpuscle and the fibrous capsule. Finally, the late contrast enhancement appears because of the poor intracapsular vascularization. This poor vascularization is due to the fibrous capsule protective barrier. Mass with contrast enhancement could differentiate the granuloma from the intracapsular seroma. When the three SIGBIC diagnosis criteria were met, we could predict GB in all cases. Adopting this diagnosis criteria, we did not have false-positive results.

We found a statistically significant association between GB and some of the equivocal BMRI BI-RADS lexicon features. Capsular contracture, water droplets, enlarged intramammary lymph node, pericapsular edema, and intracapsular seroma were associated with GB. When used the Hosmer-Lemeshow test and pseudo-R2 to associate the presence of GB with only the equivocal BMRI features, we found good sensitivity, specificity, and diagnostic accuracy. These findings could support the hypothesis that GB is underdiagnosed in our clinical practice, where most of the articles state that it is a rare event.

Analyzing the ROC curve adapting the equivocal findings proposed by the BI-RADS lexicon, we observed a good diagnostic accuracy (ROC AUC = 0.750). However, if we adopt the three specific criteria for the SIGBIC diagnosis proposed by our study, we can determine GB in 100% of patients. We believe that the proposed criteria could be incorporated into the BI-RADS lexicon to perform the diagnosis of GB with better diagnostic performance.

We hypothesized that the water droplets finding are related to permeability loss in the breast implant shell. When intracapsular fluid enters through the implant shell, a chemical reaction occurs with the inside silicone content or the silicone-made surface of the implant, which leads to silicone residues into the intracapsular compartment. Water-droplet corresponds to a macroscopic diagnosis, where the naked eye can perform the diagnosis by BMRI. We speculated that the water-droplets diagnosis by BMRI might be underestimated if compared with microscopy. (Fig 7) These findings were similar to those described as: *"breast implants, from clear to cloudy"* [23,24,25], where the authors report color changes of intact breast implants due to a chemical reaction with intracapsular fluid. In our study, when water droplets are present, the risk of SIGBIC is increased in 2.82.

We observed a high prevalence of GB in our study. Interestingly, no previous study regarding BMRI silicone implant findings has reported this occurrence neither the relation to GB. There may be different reasons for this. First, culturally most BMRI protocols for breast implant evaluation are performed without the use of contrast media. This factor may lead to a misdiagnosis of SIGBIC as a late seroma. Second, there may be a lack of diagnostic expertise. For example, before 2017, our service had not reported a positive diagnosis of SIGBIC. However, some patients were diagnosed with SIGBIC during the study period, following persistent clinical complaints, and when comparing to previous exams, we could retrospectively diagnose SIGBIC.

The two excluded patients in our study were patients with clinical complaints related to breast implants with a previous BMRI without abnormalities reported. The patients underwent a new BMRI scan in our service, where we perform the diagnosis of SIGBIC. Since our study was to establish the prevalence of SIGBIC in patients who performed the BMRI scan in a

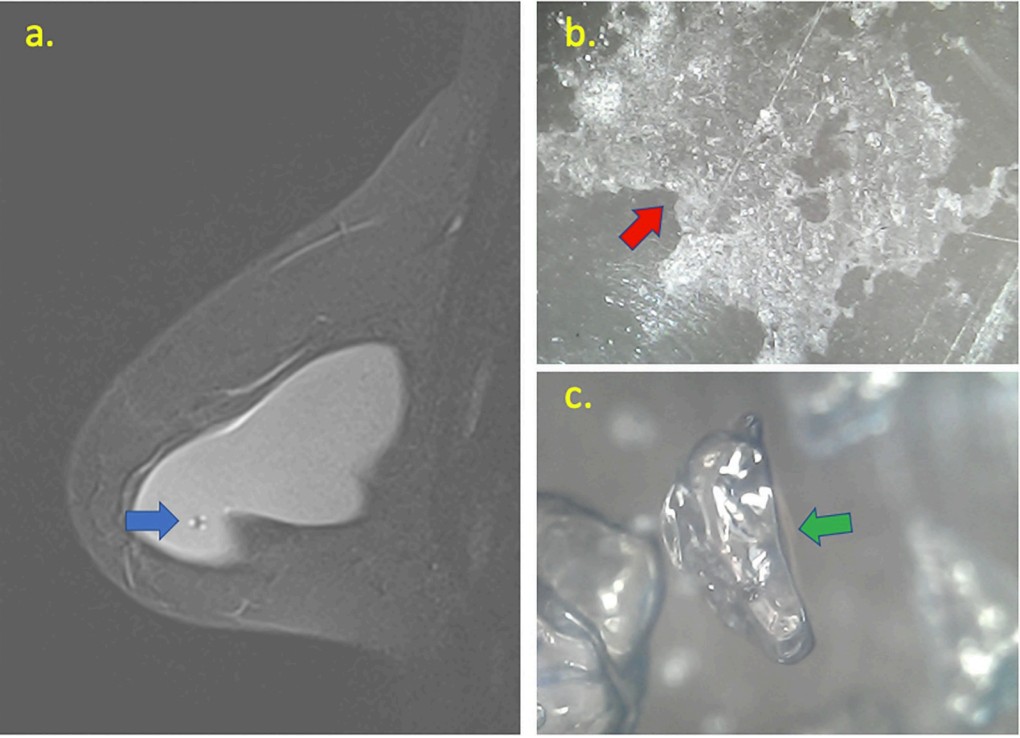

**Fig 7. Correlation between water droplets at BMRI and microscopy of implant shell.** At BMRI (a) is possible to see a focal signal change inside the breast implant pointed by the blue arrow. At the microscopy evaluation of implant shell (b) is possible to note silicone leakage pointed by the red arrow. Inside the implant with a 1.600-fold microscopy (c) it is possible to see the water droplets formed from a chemical reaction between the intracapsular fluid and the silicone gel content.

prospective study, we opted to exclude patients with previous scans forwarded to a new BMRI scan.

The relevance of this manuscript is the novelty of the issue and to alert to the possibility of gel bleeding as a precursor to complications inherent to silicone breast implants. The results exhibit a high frequency of gel bleeding in this study when adopting the restricted proposed criteria for diagnosis.

Currently, patients with clinical complaints related to breast implants, either had a diagnosis of BIA-ALCL or were considered to be an idiopathic evolutionary change without a specific causal factor. The SIGBIC diagnosis impacts patient care is in light of the high prevalence of these findings in our society. The SIGBIC diagnosis helped the patients to choose the best management of the disease focused on the trigger point of its development. Notably, the diagnosis discards the psychological origin of the symptoms which haunted many of these patients. Many patients in our clinical practice opted to breast explantation that evolves with clinical complaints remission.

This study has limitations. First, we only evaluated diagnostic BMRI in a single center. Future prospective multicenter studies should be conducted to confirm our findings. We are initiating a multicenter study involving three different breast diagnostic centers to validate our results. An additional limiting factor was that only patients diagnosed with SIGBIC underwent percutaneous biopsies or surgical capsulectomies. Although, all patients had clinical indications for the procedure. As an observational study, it was not accepted by our ethical committee to have a control group submitted to breast biopsy. However, we were able to illustrate that

MRI can diagnose GB, the novelty of this study. Future studies should access the histology of patients with capsular contracture to search for free silicone in the fibrous capsule and correlate to clinical complaints. It is essential to emphasize the importance of pathologist and radiologist training to diagnose SIGBIC.

Our study supports that the three irrevocable SIGBIC criteria proposed by the authors could predict GB in the BMRI scan. It also supports that some equivocal BMRI features proposed by the BI-RADS lexicon are related to GB. Based on our findings, we suppose GB is underdiagnosed in clinical practice and could explain most of the silicone implant reported complications.

## Supporting information

**S1 File.**
(PDF)

## Author Contributions

**Conceptualization:** Eduardo de Faria Castro Fleury.

**Data curation:** Eduardo de Faria Castro Fleury.

**Formal analysis:** Eduardo de Faria Castro Fleury.

**Funding acquisition:** Eduardo de Faria Castro Fleury.

**Investigation:** Eduardo de Faria Castro Fleury.

**Methodology:** Eduardo de Faria Castro Fleury.

**Project administration:** Eduardo de Faria Castro Fleury.

**Resources:** Eduardo de Faria Castro Fleury.

**Software:** Eduardo de Faria Castro Fleury.

**Supervision:** Eduardo de Faria Castro Fleury.

**Validation:** Eduardo de Faria Castro Fleury.

**Visualization:** Eduardo de Faria Castro Fleury.

**Writing – original draft:** Eduardo de Faria Castro Fleury.

**Writing – review & editing:** Eduardo de Faria Castro Fleury.

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
