## [Decision Letter · Decision Letter 0]

27 Apr 2020

PONE-D-20-00357

Silicone Induced Granuloma of Breast Implant Capsule (SIGBIC) diagnosis: Breast Magnetic Resonance (BMR) ability to detect silicone bleeding.

PLOS ONE

Dear Dr Fleury,

Thank you for submitting your manuscript to PLOS ONE. After careful consideration, we feel that it has merit but does not fully meet PLOS ONE’s publication criteria as it currently stands. Therefore, we invite you to submit a revised version of the manuscript that addresses the points raised during the review process.

We would appreciate receiving your revised manuscript by Jun 01 2020 11:59PM. To enhance the reproducibility of your results, we recommend that if applicable you deposit your laboratory protocols in protocols.io, where a protocol can be assigned its own identifier (DOI) such that it can be cited independently in the future. For instructions see: http://journals.plos.org/plosone/s/submission-guidelines#loc-laboratory-protocols

We look forward to receiving your revised manuscript.

Kind regards,

Pascal A. T. Baltzer, M.D.

Academic Editor

PLOS ONE

Journal Requirements:

1. In the ethics statement in the Methods and online submission information, please specify the type of informed consent that was obtained from the participants (for instance, written or verbal, and if verbal, how it was documented and witnessed).

Reviewers' comments:

Reviewer's Responses to Questions

**Comments to the Author**

1. Is the manuscript technically sound, and do the data support the conclusions?

Reviewer #1: Partly

Reviewer #2: Partly

2. Has the statistical analysis been performed appropriately and rigorously? 

Reviewer #1: No

Reviewer #2: Yes

3. Have the authors made all data underlying the findings in their manuscript fully available?

Reviewer #1: No

Reviewer #2: Yes

4. Is the manuscript presented in an intelligible fashion and written in standard English?

Reviewer #1: No

Reviewer #2: Yes

5. Review Comments to the Author

Reviewer #1: General impression:

the topic of this manuscript is relevant and is approached with a strong clinical focus. It is a niche rather than a mainstream topic. The authors are preparing their material very comprehensively. They can rely on a very large expertise and number of patients.

I am convinced that the material and the core message are worthy of publication. However, in my opinion this is difficult to do in the current form of the manuscript.

First of all, this is due to the language style: there are many difficult to understand phrases.

In addition, the structure often does not follow the established criteria of such studies. For example, the rational is only defined late in the material and methods section. Explanations that belong in the latter are presented in results and so on. a restructuring and rewriting according to STARD criteria is strongly recommended.

In addition, I think the analysis of the data is too complex and the presentation of the results is hardly comprehensible. In the end, the statement is simple. Can we make a reliable diagnosis with additional diagnostic criteria? If so, how exactly? On the other hand, there are too many and too complex analyses, the details even appear to be erroneous (percentages Table 1) and in some cases are not necessary to convey the message of the authors. More important details such as the multivariate model and the ROC curve, however, are not mentioned at all.

Detailed comments

The abstract should be shortened . The methods have to be better defined here and the rational has to be better explained.

In the M&M, the setting of the hospital according to the STARD criteria should be better explained.

A clear description of the mri protocol is missing

I find the image material well prepared, however, on page 14, line 116 ff. the features should be supplemented with an example image + schematic drawing

Examples of incomprehensible wording can be found on page 15, lines 198 and 199

I do not consider it appropriate to set alpha at 25%.

in order to better examine the accuracy of the criteria, I suggest that patients +/- SICBIG to be analysed more consistently

The three patients who had to be excluded should be described in more detail

confusing is the moderate accuracy of multivariate models in the context of the otherwise very optimistic univariate results

Reviewer #2: The paper is nicely written but I would suggest to better emphasize what should be the clinical consequence of these findings. I also find quite difficult to justify that the authors performed percutaneous breast biopsy or even capsulectomy for no suspicious findings, especially considering the risk to damage the implant. I would also suggest to improve the figure descriptions that are quite crucial to understand what the authors refer to.

Introduction

“However, gel bleeding is virtually ignored by the academy as a trigger point for these changes. Most articles advocate that gel bleeding is a rare event with no pathological significance. There are no studies in the literature demonstrating the frequency of this event in BMRI scans. “

I would rephrase this sentence. Gel bleeding is not virtually ignored but, as far as I know, there are indeed no demonstrations that it can determine any of the known complications.

M&M

How could you justify the biopsy or even the capsulectomy of these no suspicious findings also considering the not negligible risk to damage the implant? please specific better the criteria

EILN: I guess 0.5 cm is a quite low cut-off especially in the axillary/ periprosthetic tissue. Please comment.

Results

Fig. 2 Could you find a sonographic finding correlating to the MRI finding?

Fig.3 What is the curve referring to? Please describe more in detail the figures.

Fig.5 Please report separately the two cases and describe more in detail the figures (MRI sequence, finding, etc.)

6. PLOS authors have the option to publish the peer review history of their article (what does this mean?). If published, this will include your full peer review and any attached files.

Reviewer #1: No

Reviewer #2: No

---

## [Author Response · Author response to Decision Letter 0]

4 May 2020

PONE-D-20-00357

Silicone Induced Granuloma of Breast Implant Capsule (SIGBIC) diagnosis: Breast Magnetic Resonance (BMR) ability to detect silicone bleeding.

PLOS ONE

Dear Dr Fleury,

Thank you for submitting your manuscript to PLOS ONE. After careful consideration, we feel that it has merit but does not fully meet PLOS ONE’s publication criteria as it currently stands. Therefore, we invite you to submit a revised version of the manuscript that addresses the points raised during the review process.

We would appreciate receiving your revised manuscript by Jun 01 2020 11:59PM. To enhance the reproducibility of your results, we recommend that if applicable you deposit your laboratory protocols in protocols.io, where a protocol can be assigned its own identifier (DOI) such that it can be cited independently in the future. For instructions see: http://journals.plos.org/plosone/s/submission-guidelines#loc-laboratory-protocols

ADDED

• A rebuttal letter that responds to each point raised by the academic editor and reviewer(s). This letter should be uploaded as separate file and labeled 'Response to Reviewers'.

• A marked-up copy of your manuscript that highlights changes made to the original version. This file should be uploaded as separate file and labeled 'Revised Manuscript with Track Changes'.

• An unmarked version of your revised paper without tracked changes. This file should be uploaded as separate file and labeled 'Manuscript'.

We look forward to receiving your revised manuscript.

Kind regards,

Pascal A. T. Baltzer, M.D.

Academic Editor

PLOS ONE

Journal Requirements:

1. In the ethics statement in the Methods and online submission information, please specify the type of informed consent that was obtained from the participants (for instance, written or verbal, and if verbal, how it was documented and witnessed). OK

Reviewers' comments:

Reviewer's Responses to Questions

Comments to the Author

1. Is the manuscript technically sound, and do the data support the conclusions?

Reviewer #1: Partly

Reviewer #2: Partly

Hope the new version is adequate for publishing in PLOS ONE.

2. Has the statistical analysis been performed appropriately and rigorously?

Reviewer #1: No

Reviewer #2: Yes

Revised the methodology.

3. Have the authors made all data underlying the findings in their manuscript fully available?

Reviewer #1: No

Reviewer #2: Yes

Added to protocols.io

dx.doi.org/10.17504/protocols.io.bfn4jmgw

4. Is the manuscript presented in an intelligible fashion and written in standard English?

Reviewer #1: No

Reviewer #2: Yes

Revised

5. Review Comments to the Author

Reviewer #1: General impression:

the topic of this manuscript is relevant and is approached with a strong clinical focus. It is a niche rather than a mainstream topic. The authors are preparing their material very comprehensively. They can rely on a very large expertise and number of patients.

I am convinced that the material and the core message are worthy of publication. However, in my opinion this is difficult to do in the current form of the manuscript.

First of all, this is due to the language style: there are many difficult to understand phrases.

In addition, the structure often does not follow the established criteria of such studies. For example, the rational is only defined late in the material and methods section. Explanations that belong in the latter are presented in results and so on. a restructuring and rewriting according to STARD criteria is strongly recommended.

In addition, I think the analysis of the data is too complex and the presentation of the results is hardly comprehensible. In the end, the statement is simple. Can we make a reliable diagnosis with additional diagnostic criteria? If so, how exactly? On the other hand, there are too many and too complex analyses, the details even appear to be erroneous (percentages Table 1) and in some cases are not necessary to convey the message of the authors. More important details such as the multivariate model and the ROC curve, however, are not mentioned at all.

Dear revisor, Thank you for the very important commentaries. I am convinced they improved the quality of the manuscript and made the manuscript more readable. Hope in this new version it fits the criteria for publishing in PLOS ONE.

I followed the STARD 2015 protocol.

Detailed comments

The abstract should be shortened . The methods have to be better defined here and the rational has to be better explained.

Rewritten

In the M&M, the setting of the hospital according to the STARD criteria should be better explained.

Explained

A clear description of the mri protocol is missing

This is the standard MRI protocol, available at protocols.io (dx.doi.org/10.17504/protocols.io.bfn4jmgw)

I find the image material well prepared, however, on page 14, line 116 ff. the features should be supplemented with an example image + schematic drawing Added.

Examples of incomprehensible wording can be found on page 15, lines 198 and 199

Excluded

I do not consider it appropriate to set alpha at 25%.

I QUESTIONED THE STATISTICS AND HE EXPLAINED ME TO ADOPT THIS APLHA AS IS CITED IN REFERENCES.

in order to better examine the accuracy of the criteria, I suggest that patients +/- SICBIG to be analysed more consistently

The three patients who had to be excluded should be described in more detail

EXPLAINED IN DISCUSSION

confusing is the moderate accuracy of multivariate models in the context of the otherwise very optimistic univariate results

THE ACCURACY IS MODERATE/ GOOD BECAUSE THE EQUIVOCAL FINDINGS WHERE NOT SPECIFIC FOR GB. WE ADOPTED THE CURRENT BI-RADS DESCRIPTORS TO PREDICT GB

Reviewer #2: The paper is nicely written but I would suggest to better emphasize what should be the clinical consequence of these findings. I also find quite difficult to justify that the authors performed percutaneous breast biopsy or even capsulectomy for no suspicious findings, especially considering the risk to damage the implant. I would also suggest to improve the figure descriptions that are quite crucial to understand what the authors refer to.

THANK YOU FOR THE COMMENTS. THEY CONTRIBUTE TO ENHANCE THE POWER OF MY MANUSCRIPT.

Introduction

“However, gel bleeding is virtually ignored by the academy as a trigger point for these changes. Most articles advocate that gel bleeding is a rare event with no pathological significance. There are no studies in the literature demonstrating the frequency of this event in BMRI scans. “

I would rephrase this sentence. Gel bleeding is not virtually ignored but, as far as I know, there are indeed no demonstrations that it can determine any of the known complications. 

PERFECT IT IMPROVES THE SOUNDNESS OF THE ARTICLE

M&M

How could you justify the biopsy or even the capsulectomy of these no suspicious findings also considering the not negligible risk to damage the implant? please specific better the criteria. 

ADDED PATIENTS WITH SPECIFIC SYMPTOMS. WE ALSO REPORT THAT WE DIDN’T HAVE ANY COMPLICATION REGARDING BIOPSY.

EILN: I guess 0.5 cm is a quite low cut-off especially in the axillary/ periprosthetic tissue. Please comment. 

WE ADOPTED FOR INTRAMAMMARY LYMPH NODE THE CUT OF OF 0.5CM IN THE PERPENDICULAR AXIS, WE ADDED THIS INFORMATION TO THE MANUSCRIPT.

Results

Fig. 2 Could you find a sonographic finding correlating to the MRI finding?

YES. IN THIS ARTICLE WE DESCRIBED SOME ULTRASOUND FEATURES. I described some specific features in figures 4 and 5.

Application of Breast Ultrasound Elastography to Differentiate Intracapsular Collection from Silicone-Induced Granuloma of Breast Implant Capsule Complementarily to Contrast-Enhanced Breast Magnetic Resonance Imaging

Fig.3 What is the curve referring to? Please describe more in detail the figures. 

OK

Fig.5 Please report separately the two cases and describe more in detail the figures (MRI sequence, finding, etc.)

OK

---

## [Editor Report · Decision Letter 1]

9 Jun 2020

Silicone Induced Granuloma of Breast Implant Capsule (SIGBIC) diagnosis: Breast Magnetic Resonance (BMR) sensitivity to detect silicone bleeding.

PONE-D-20-00357R1

Dear Dr. Fleury,

Dear colleague!

We are pleased to inform you that your manuscript has been judged scientifically suitable for publication and will be formally accepted for publication once it complies with all outstanding technical requirements.

With kind regards,

Pascal A. T. Baltzer, M.D.

Academic Editor

PLOS ONE

Additional Editor Comments (optional):

Please do very carefully proofread your article. Being not a native english speaker myself, I understand that the manuscript is not free of small typographic and idiomatic mistakes, but there are some ackward typos (pos instead of post in a figure, some pseudo-english technical terms) I would strongly suggest you to correct in the prior to production.
---

## [Editor Report · Acceptance letter]

17 Jun 2020

PONE-D-20-00357R1 

Silicone Induced Granuloma of Breast Implant Capsule (SIGBIC) diagnosis: Breast Magnetic Resonance (BMR) sensitivity to detect silicone bleeding. 

Dear Dr. Fleury:

I'm pleased to inform you that your manuscript has been deemed suitable for publication in PLOS ONE. Congratulations! Your manuscript is now with our production department. 

Kind regards, 

on behalf of

Dr. Pascal A. T. Baltzer 

Academic Editor

PLOS ONE